# An Instrumented Golden Eagle’s (*Aquila chrysaetos)* Long-Distance Flight Behavior

**DOI:** 10.3390/ani12111470

**Published:** 2022-06-06

**Authors:** Michael Garstang, Steven Greco, George D. Emmitt, Tricia A. Miller, Michael Lanzone

**Affiliations:** 1Environmental Sciences and Simpson Weather Associates, Inc., University of Virginia, Charlottesville, VA 22904, USA; 2Simpson Weather Associates, Inc., Charlottesville, VA 22902, USA; sxg@swa.com (S.G.); gde@swa.com (G.D.E.); 3Conservation Science Global, West Cape May, NJ 08204, USA; trish.miller@consciglobal.org; 4Cell Track Tech, Rio Grande, NJ 08242, USA; michael.lazone@celltracktech.com

**Keywords:** migration, bird flight, atmospheric waves, energy consumption

## Abstract

**Simple Summary:**

All large birds have devised ways to save on the costly energy demands of flight. Geese, which form into a precise V formation, are a familiar example. Current measurements from GPS instrumentation attached to birds provide the exact location of the bird at every second of its flight. These measurements are transmitted during migration. An analysis of such data for a single 105 km (~70 mile) segment of a golden eagle’s flight illustrates how the bird makes use of atmospheric waves near 2000 m above the ground to repetitively climb 100s of meters while circling, followed by long glides to advance along its migratory route. From this height, in sixteen circling/gliding episodes, the eagle covers more than 100 km by harvesting atmospheric wave energy. Such details of a single bird’s 2 h segment of flight provide insight into how a soaring bird can cross continents and oceans, and even rise over the world’s highest mountains.

**Abstract:**

One-second-processed three-dimensional position observations transmitted from an instrumented golden eagle were used to determine the detailed long-range flight behavior of the bird. Once elevated from the surface, the eagle systematically used atmospheric gravity waves, first to gain altitude, and then, in multiple sequential glides, to cover over 100 km with a minimum expenditure of its metabolic energy.

## 1. Introduction

The dependence of large birds, such as eagles, on their wings during long-distance flight in atmospheric fields of gravity, stability, and motion, including simply remaining airborne, represents a significant drain upon their reserves of metabolic energy. Once at altitude, these birds are adept at converting the geopotential energy that is gained during climbs to effective gains in distance [1,2,3,4].

The use of atmospheric features by large birds in migration has been the focus of extensive research over the last fifty years. Previous studies have identified the vertical motions that are generated by surface-based thermals [5,6,7,8,9,10,11] as the most favored by birds for climbing and soaring over land. Dynamic soaring (responding to the vertical shear of the horizontal wind [12,13,14,15,16]) is another atmospheric feature that is used by the albatross to maintain prolonged flight over the ocean. Orographic uplift or slope (soaring at a low altitude over complex terrain) has also been evaluated [8,10,17,18]. Birds have been known to use the inflow region beneath the convective cloud base to gain altitude [19] in the lowest levels of the atmosphere.

Over the ocean, roll vortices and near-surface velocity-convergence lines create the upward velocity fields used by birds [12,20]. At higher elevations, gravity waves can be initiated by deep convective cloud systems, with upward motions that reach the tropopause in a stable atmosphere, which initiate a gravity wave [21]. Most, if not all, surface-initiated vertical velocities in the atmosphere decay rapidly with height (<1000 m) or are modified by the condensation of water vapor, the formation of cloud and rain, or the release of latent heat [21]. The experiences of glider pilots have also been usefully drawn upon to describe bird flight and, in particular, to explain how birds gain initial height from their earth-bound habitat to reach near and distant locations [10,22,23]. In the course of extended migrations under varying atmospheric conditions and surface topography, the bird may use more than one of these modes of flight subsidy [5,24,25].

Birds have been shown to reach heights near to 10,000 m over the Himalayas and the Andes, and close to these heights over the open oceans [16,26,27,28]. A few early studies identified standing atmospheric waves that were used by large birds to climb to and fly at considerable altitudes [18,29,30,31,32]. However, these studies were performed without the benefit of either high-resolution Global-Positioning-System (GPS) data or fine-scale meteorological models to describe the atmospheric motions.

In this paper, we focus on internal atmospheric gravity waves that occur within stable layers embedded in the troposphere. Within these layers, the density decreases with height, providing a restoring force (gravity) to vertical motions initiated by mechanisms such as cloud convection, individual mountain ridges, and at transitions from areas of low surface roughness to areas of high roughness. The result is a train of waves that are referred to as internal gravity waves. These waves have amplitudes, wavelengths, and frequencies that are determined, in large part, by the density gradient within the stable layer. Such wave structures display recognizable features of wavelength, height, and amplitude, which are reflected by the migratory flight of the instrumented eagle discussed in this paper.

We used high-time-resolution position measurements from an instrumented golden eagle (*Aquila chrysaetos*) to conduct a detailed analysis on how this bird repeatedly uses gravity waves to gain and convert the altitude to horizontal distance while maintaining a consistent migratory heading. These results show that, over this day’s flight segment, the eagle generated sufficient energy from the atmosphere to support the flight distance covered. Such high-volume high-resolution in situ avian measurements, combined with high-resolution numerical models, provide the framework for advancing our understanding of soaring birds’ use of internal gravity waves during migration.

## 2. Materials and Methods

### 2.1. Instrumentation and Data

We trapped an adult female golden eagle weighing 6100 g with a bow net over bait in North Carolina. Once in hand, the bird was banded with an aluminum United States Geological Survey (USGS) leg band and outfitted with a GPS–GSM (Global Positioning System—Global System for Mobile Communications) telemetry device (Cellular Tracking Technologies, LLC, Rio Grande, NJ, USA). For this analysis, we reprogrammed the unit to collect data at 15 min intervals when the bird was not moving, and at ~1 sec intervals when the bird was moving. The published accuracy of the GPS collar is <2m horizontal and <22.5 m vertical (19). The telemetry device was attached in a backpack style [33] using a Teflon ^®^ ribbon (Bally Ribbon Mills, Bally, PA, USA). Animal trapping was permitted by the state of North Carolina (Permit # 13-BB00064), the Bird Banding Lab of the US Geological Survey (Permit #23715), and an animal-care-and-use protocol from West Virginia University (14_0303). The GPS collar data are expressed in meters above sea level (ASL). For each flight point, the terrain elevation was determined from colocated (latitude/longitude) United States Geological Survey (USGS) digital-elevation-model (DEM) data within the Google Earth API. The eagle’s height above ground level (AGL) was derived from the difference between the ASL height and the terrain height.

### 2.2. Flight Description 

We analyzed a segment of the eagle’s flight starting on 24 March 2016 at 15:57:46 UTC (11:57:46 LST), SW of Bergton, VA, in the eastern Appalachian Mountains, and ending at 18:04:34 UTC near Cumberland, MD. The eagle flew on an NNE heading of ~20 degrees, at an average surface speed of 16.5 m s^−1^, over an actual path distance of 125.5 km, and a direct linear (migration) distance of 103.5 km (Figure 1). Figure 1 inserts show that the topography was characterized by secondary orthogonal ridges within the primary Appalachian orientation. In Figure 2, the entire flight path is subdivided into eight 12–15 km segments, with Segment 1 being the start of the SSW–NNE flight. Sixteen yellow ovals mark the areas where the eagle used 3 or more consecutive circles to gain altitude. Figure 3a shows the detailed changes in the height of this flight relative to the underlying terrain. Figure 3b provides an expanded view of two sequential circling maneuvers as the eagle climbed to 1300 m AGL.

### 2.3. Meteorological Data

The National Oceanic and Atmospheric Administration (NOAA) Geostationary Operational Environmental Satellite (GOES) imagery and NOAA’s National Weather Service (NWS) surface weather maps were utilized. In addition, NWS surface observations for Grant County, WV, and Cumberland, MD, and upper air observations from Dulles International (KIAD), were used in this study.

### 2.4. Atmospheric Model and Data 

The meteorological-community open-source Weather Research and Forecasting (WRF) model, primarily developed by the National Center for Atmospheric Research (NCAR) [34], was used to provide local and larger-scale circulation fields in and above the boundary layer, as well as comparative vertical velocities to those calculated from the high-resolution measurements of the eagle-flight instrument package. 

The features and parameterization schemes of the WRF model, specifically chosen for the complex terrain in this study, are provided in Table 1. We used the WRF model with three nested domains (9 km, 3 km, and 1 km horizontal resolutions), with a vertical resolution available of <100 m between the surface and 3 km, while mainly focusing on the results for the 1 km domain to describe local and larger-scale circulations in and above the boundary layer.

### 2.5. Weather Conditions during Flight 

NOAA GOES cloud imagery at 1730 UTC on 24 March 2016 (Appendix A), and the NWS surface-weather-report map (Appendix A) and observations (Appendix A), showed largely clear sunny conditions, with surface winds of about 10 mi h^−1^ (4.5 m s^−1^) out of the S–SW at the beginning of the eagle flight, and veering to the SSE at the end of the flight. Upper air observations from KIAD (Appendix A) show winds at 750 m at 12 UTC, 24 March (before the flight), and at 00 UTC on 25 March (after the flight), when it changed from light and variable to 20 mi h^−1^ (9.0 m s^−1^), while the winds at the eagle flight level changed from W at 20 mi h^−1^ (9 m s^−1^) to SW–S at 25 mi h^−1^ (11.5 m s^−1^). The stable layer between 775 and 700 mb in Appendix A is located at the general altitude (2 km ASL) of the eagle’s circle and the glide migratory flight within an internal-gravity-wave train.

### 2.6. Flight-Maneuver Classification

The high-resolution GPS data were first manually compiled in a cartesian coordinate framework to display and identify three primary flight maneuvers: circling, gliding, and meandering. The GPS ~1.0 sec data were smoothed with a linear 3-value running mean to reduce high-frequency variability in the position data.

The positional data were converted to vertical and horizontal velocities, including true air speed (TAS) and ground speed (GS). The eagle’s TAS is the speed of the eagle relative to the air through which it is flying and, thus, is directly related to the lift being generated by its wings. TAS is independent of the ambient wind speeds, which are relative to the ground. Thus, the GS of the bird is the vector sum of its TAS and the wind speed at its flight level.

We performed our analyses of vertical ambient air motions from the GPS data making three assumptions: (a) The eagle migrates without flapping its wings, except for its starting takeoff. While not specific to the golden eagle, this assumption is supported by studies of GPS-instrumented Andean condors, with the finding that the condor flapped its wings for only 0.8% of the migration flight [35]; (b) The eagle did not dive, especially at or near 2000 m AGL. Any rapid and short-lived descents were likely due to turbulence-scale (<10 m) vertical velocities [36]; and (c) The eagle is migrating and not foraging.

Circling was defined as near-circular flight, with the eagle returning to its original heading within ±10 degrees of azimuth, before continuing with linear flight or entering another circle. Occasionally, rotational directional changes occurred in consecutive circles from clockwise to counterclockwise, or vice versa (Figure 3b). 

Linear gliding flight closely followed the heading of the migration path and was defined as any flight segment where heading did not change more than 10 degrees in 20 s. We used the conservative value of −0.75 m s^−1^ for the golden eagle’s best-glide-speed sink rate for non-turning flight, independent of any atmospheric vertical lift. Other research has reported best-glide-speed sink rates between −0.94 m s^−1^ and −1.9 m s^−1^ [19] and −0.9 m s^−1^ and −1.33 m s^−1^ [4] for vultures. However, during circling, we assumed a 25-degree bank angle, which reduced the bird’s vertical lift by ~10%, and thus the sink rate increased to ~−0.82 m s^−1^. Given the published higher sink rates, our values for the golden eagle may result in an underestimation of the computed vertical atmospheric wind vector needed to explain the eagle’s GPS-measured vertical motion. Upward glides were defined by climb rates greater than 0.75 ms^−1^ to offset the assumed sink rate of −0.75 m s^−1^ at the best glide speed. All other glide segments were classified as downward.

Meandering was characterized by significant heading changes by the eagle without closing a circle, possibly serving as a searching pattern for atmospheric vertical motions (lower left corner of Insert 1 in Figure 1).

## 3. Results

### 3.1. Small-Scale Flight Behavior

Three distinct flight characteristics are described above:Circling (Figure 1, Figure 2 and Figure 3b);Linear gliding (Figure 2);Meandering (Figure 1).

The bird executed 86 circles, with approximate diameters between 30 and 40 m, and an average duration of 22 s, occupying ~26% of the flight time in this 2 h segment of the flight migration (Table 2, Table 3 and Appendix A). In 85 of these circles, the eagle gained altitude. The eagle only lost altitude (~6 m) in a single circle. The average gain in altitude in each positive (upward) circle was ~37 m, with a maximum gain within a single circle of +70 m. The eagle had a cumulative gain of +3227 m in altitude generated in 85 positive circles, compared to a gain of +401 m and a loss of −4394 m, respectively, in meanders and glides (Table 2).

The eagle climbed to ~1350 m AGL in the initial 29 consecutive circles, over a time period of 1000 s (Figure 3). The eagle’s climb rates exceeded the bird’s best-glide sink rate of −0.75 m s^−1^ in all circles and increased with altitude, as shown in Appendix A. The height gains associated with each of the circles in this initial climb also increased with height and ranged between 0 and 60 m (Appendix A). 

The TAS of the eagle was assumed to be nearly constant within each circle and was determined from the amplitude of the sine wave in the ground speed (Appendix A). The TAS of the eagle circling near 1460 m AGL, or 1950 m above sea level (ASL), was determined to be 11.4 m s^−1^ (Appendix A). At a lower altitude of 480 m AGL, or 980 m ASL, the TAS within a circle was determined to be 4.5 m s^−1^.

Using the TAS and GS, we derived an estimate of the horizontal wind speed within several vertical layers of the atmosphere traversed by the eagle circling during the initial climb. At any point in a circle, the ground speed of the eagle is the vector sum of the ambient wind speed and the bird’s TAS. The ambient wind speed was determined by the average ground speed during a circling maneuver. The ambient winds (Appendix A) derived from the eagle’s circling maneuvers were consistent with the measured wind soundings observed by the KIAD rawinsonde (Appendix A). 

### 3.2. Atmospheric Waves

The WRF model was used to examine the velocity fields through which the eagle flew during this migration. The vertical velocity fields for Domain 3 (1 km resolution) and Domain 2 (3 km resolution) are shown in Figure 4. At 250 m AGL for Domain 3 (Figure 4a), the model showed a SW–NE organized structure, suggesting local terrain-induced waves near the surface, with vertical velocities up to ±1.0 m s^−1^, while Figure 4b,c showed alternating vertical motions, indicative of atmospheric waves that are generated at altitude (1750 m AGL) for both the 3 km and 1 km resolutions. These waves are perpendicular to the major spines of the mountain ridges and to the direction of the prevailing wind and the eagle’s flight track. Vertical cross sections of the WRF-model Domain-3 wind speed, wind direction, and vertical velocity along the flight path of the eagle (white track in Figure 4) were also generated and are shown in Appendix A.

Limitations of the single (1700 UCT) 100 m-resolution snapshot estimate of the modeled wind field provides a supportive but not definitive comparison to the flight results determined from the bird-borne instrumentation. The model results clearly show the existence of alternating wave-induced vertical velocities near the eagle flight level.

### 3.3. Eagle’s Climb and Glide Speeds

The eagle’s climb rate within these waves increased with altitude (Appendix A) during its initial climb to 1.35 km AGL, with an average altitude gain per circle of 30.6 m (Appendix A). This increase in the climb rate with altitude is consistent with strong upward motion within atmospheric gravity waves.

Appendix A presents the eagle’s movement and speeds following the two-step initial rise. After this climb, the eagle’s altitude oscillated between 600 m and 2000 m AGL, with climb rates ranging between 6.1 and 8.2 m s^−1^ (Appendix A). The ground speeds (GS) over the 125.5 km migration path varied from near zero meters per sec to over 30 m s^−1^ (Appendix A). The eagle’s vertical velocities and associated glide speeds were consistent with an assumed best-glide-slope “no flapping” sink rate of −0.75 m s^−1^. Once at the general migration level of ~1350 m AGL (Figure 3a), downward gliding flight produced an average GS of ~20 m s^−1^, with a maximum GS of 34 m s^−1^ at the end of the recorded data (Appendix A). Accounting for the winds at flight level, the maximum forward speed due to the eagle’s glide was in the order of 24 m s^−1^.

The summary results shown in Table 2 were used to determine whether the accumulated height gain from circling supported the two-hour flight while conserving metabolic energy. During the total 32 min (Table 3) of the 16 primary circling sequences (Figure 2), we assumed that circling contributed no net motion towards the migration target. There was, however, forward motion due to the ambient tailwind. For the 76 min of gliding (Table 2), we used a lift/drag (L/D) ratio of 15 for large birds [37,38,39,40], together with the measured average GS of 19.1 m s^−1^, which included the tailwind and forward air speed of the eagle in the best glide configuration. The total climb and glide distance along the migration path and within the atmospheric waves was calculated to be 108.2 km, the sum of Dc and Dg (Table 3), which provides sufficient available atmospheric geopotential energy to support the eagle’s 103.5 km journey, while not requiring flapping and the consumption of its metabolic resources.

In this case, surface-based atmospheric vertical motions in the first 1000 s of the flight (Figure 1 and Figure 3a) were used by the eagle to climb to ~1350 m AGL, where gravity waves provided opportunities to climb and glide without descending to the near surface, where the bird would have needed to use flapping or find near-surface vertical motions to remain aloft.

## 4. Discussion

These detailed observations and model calculations did not find evidence of wind shear or thermals, which would be sufficient to account for the eagle’s observed climb rates and patterns of climbs and glides. Both the model wind profile and the wind profile derived from the eagle’s GPS data (Appendix A) estimated the wind-speed shears in the climb-out area to be near 0.007 s^−1^, which is insufficient to support dynamic soaring. No evidence in the WRF-model calculations supported dynamic soaring, which further suggests that this flight mechanism was not present on this day at or near 1300 m AGL along this route segment. 

During the flight of this eagle, the surface temperatures in the region were between 72 °F and 79 °F (Appendix A). The dominant vegetation below the flight track of the bird consists of near continuous midlatitude deciduous forest, with little, if any, natural open ground. Under late-winter/early-spring conditions, it is unlikely that strong surface-temperature gradients existed to initiate and support thermals. Multiple successive “hot spots”, with nearly constant spacing along a 120 km flight path, are not likely to exist, as evidenced by the observed spacing of the circling and the climb and glide patterns (Figure 1 and Figure 2). Alternating clockwise and counterclockwise circling within a contiguous series of climbing circles is further evidence in support of the slab-like vertical velocity zones that are typical of waves. The eagle’s change in direction happens intermittently during a continuous series of climbing circles. Such a reversal of flight is not probable within the continuous cylindrical rotation of an individual thermal.

Large birds are known to use atmospheric waves in migration flights. Over complex terrain, these waves are standing atmospheric internal gravity waves that result in a harmonic oscillation in stably stratified flow over an obstacle. Such waves have been seen and documented from satellite imagery over the Blue Ridge and the central Appalachians of the eastern United States [41,42], and, in classical form, in the lee of the Southern Alps of New Zealand’s South Island [43].

At altitudes well above the surface, gravity waves can orientate perpendicular to the wind, without regard to the orientation of individual mountain ridges and topography (Figure 4b,c). The more regional surface drag of the Appalachians, as a whole, results in harmonic motions that, at higher altitudes, propagate vertically and horizontally [44], and, in this case, are perpendicular to the winds at the migration altitude while being approximately parallel to the spines of the underlying Appalachian ridges. 

Our WRF-model results show the regional internal gravity waves with both the 1 km (Figure 4b) and 3 km (Figure 4c) horizonal-resolution domains. The height of the modeled waves (~2 km ASL) is consistent with the observed stable layer near 775–700 mb in the NOAA sounding (Appendix A). The 3 km grid shows a smoother and more idealized wave pattern, with weaker magnitudes and wavelengths of ~10 km. At the higher grid resolution of 1 km, the model resolves a vertical velocity field with higher vertical velocities, showing a more complex wave pattern. While the 1 km model still cannot resolve waves with wavelengths much less than 5 km, it strongly suggests nine full wave cycles, without precluding the shorter energetic waves between the beginning and end of the 2 h flight. The eagle flight data (Figure 3a) showed sixteen primary climb and glide cycles, and as expected, the model’s vertical motions were weaker in magnitude than those derived from the flight record. This pattern of vertical motions is consistent with having waves with both dominant wavelengths around 10 km, and additional energetic waves of shorter wavelengths.

Although the exact number of full wave cycles and the strength of the vertical velocities in the real atmosphere and as determined from the eagle GPS data were not totally resolved by the model, the model-generated vertical-velocity-field patterns presented in Figure 4 and Appendix A were consistent with the motions determined from the GPS data and support waves as a reasonable explanation for the eagle’s flight maneuvers during migration. It is equally important to note that atmospheric gravity waves are triggered over the open oceans in response to deep convective storms [20]. Frigate birds and raptors make long over-ocean flights that are supported by gravity waves between Africa and the Himalayas over the northern Indian Ocean [16].

## 5. Conclusions

The consistency and validity of the derived numerical results from the high temporal and spatial measurements reported from an attached instrument provide unique insight into flight behavior and serve equally to generate high-frequency time and space measurements of the atmospheric velocity fields. The precise numerical description of deliberate flight maneuvers, coupled with a pervasive atmospheric wave system, provides a tractable framework within which the interactions between this species and its flight environment can be assessed. 

The execution of repetitive circles throughout this eagle’s migratory flight is evidence that the bird used this flight maneuver to gain altitude once it entered a region of organized upward atmospheric motion.

The positive (upward) displacement of the bird in all but one of the 86 circles, together with the rates of the rise in the excess of its sink rate, which led to a significant gain in the altitude in each of the circles, is evidence of the eagle’s ability to locate, identify, and utilize atmospheric velocity fields to support upward motion.

The multiple consecutive circling (with adjacent clockwise and anticlockwise circles) in a meandering pattern suggests searching by the bird for locally induced vertical velocities to gain access to gravity waves generated by the regional drag of the Appalachian topography. Successive circling with reversing directions of circles were incompatible with the use of thermals.

The eagle’s descent rates during linear glides from higher altitudes frequently exceeded its estimated sink rate for its best-nonturning-glide ratio. This finding is consistent with organized downward motions within the troughs of atmospheric gravity waves.

Circling eighty-six times in sixteen episodes, followed by forward gliding while maintaining a constant heading towards a predetermined destination, called upon a high degree of accurate navigation. 

Including a beneficial tailwind, the accumulated height gained by the eagle’s systematic use of circling was in excess of that needed to support the necessary gain in forward motion in glides while conserving the bird’s metabolic energy. 

Perhaps the singular most important conclusion that can be drawn from this detailed analysis of a section of a golden eagle’s migrational flight is that it presents, in total, a recognizable signature that is predictive of its intent to migrate. This central conclusion provides the basis to formulate a testable hypothesis that is manifested in the details of the high-resolution measurements made by an in situ instrument system and that reflects a complex but clearly recognizable flight pattern.

The capacity to record and transmit in situ and environmental measurements at time intervals of one second and below, over global-scale distances, opens an infinity of opportunity that is rapidly becoming available to the understanding of complex and heretofore obscured flight behavior. These current and future measurements provide the opportunity to apply and expand this methodology.

## Figures and Tables

**Figure 1 animals-12-01470-f001:**
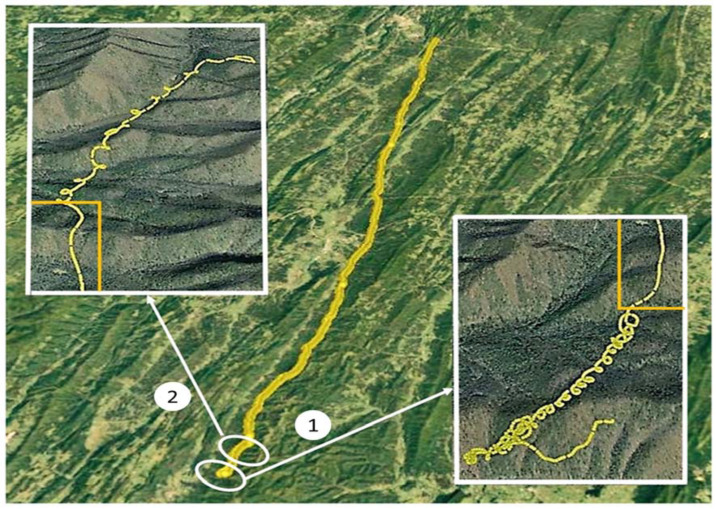
The 103.5 km flight path of the eagle is shown from near Bergton, VA, to just east of Cumberland, MD. From takeoff to the start of the first series of circles (Insert 1), the eagle demonstrates a meandering flight pattern. The bird rises to 75 m above ground level (AGL) in a series of 17 circles (Insert 1), then glides for 2 min (right and left small orange rectangles). The second series of 12 circles (Insert 2) carried the eagle to 1300 m AGL. Note that the first series of circles (labeled 1) were clockwise, while, in the second series (labeled 2), the eagle changes direction on three occasions.

**Figure 2 animals-12-01470-f002:**
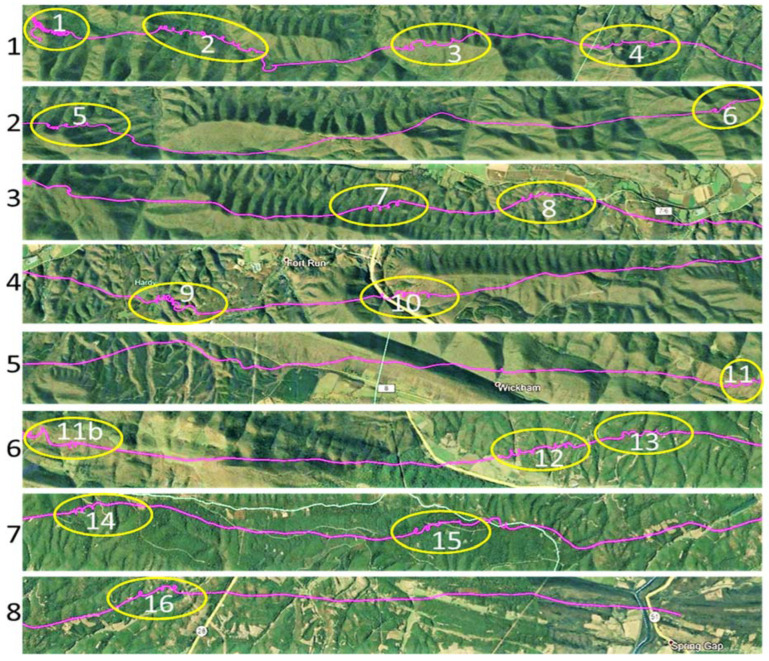
Eight 12–15 km segments, flowing continuously from left to right, and linking 16 yellow ovals marking the locations of three or more consecutive circles, collectively making up the 103 km flight path shown in Figure 1. The circling flight patterns contained within the 16 yellow ovals produce a gain in altitude, while the connecting flight segments between circle clusters consist of descending linear gliding.

**Figure 3 animals-12-01470-f003:**
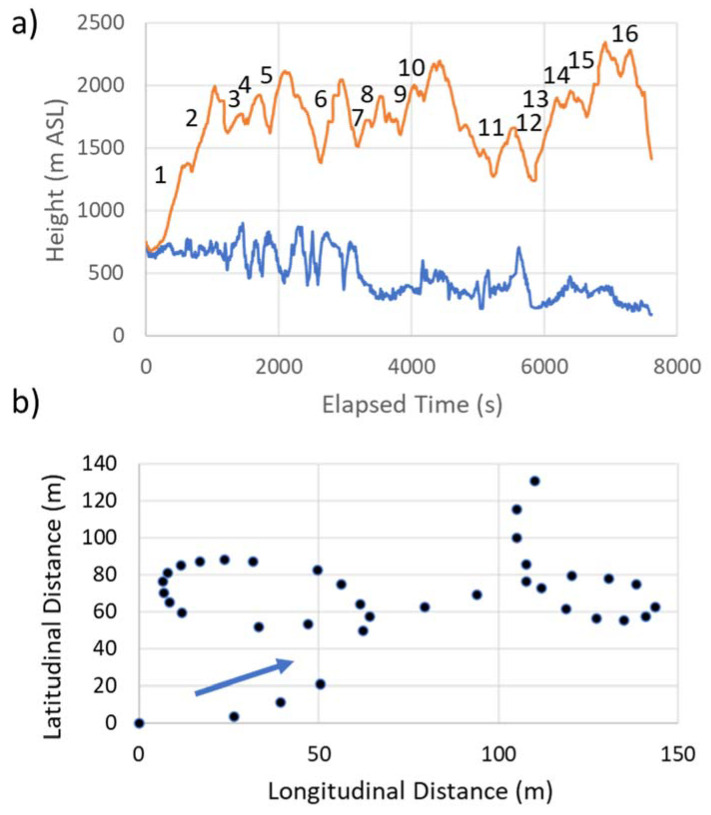
(**a**) Eagle’s flight altitude at ~1 s intervals (orange in m ASL), and height (blue in m ASL) of topography beneath the bird. Numbering along the flight path references the 16 yellow circle clusters shown in Figure 2. (**b**) Example of two sequential circles with the eagle flying in the direction of the blue arrow. The black dots are ~1 s apart.

**Figure 4 animals-12-01470-f004:**
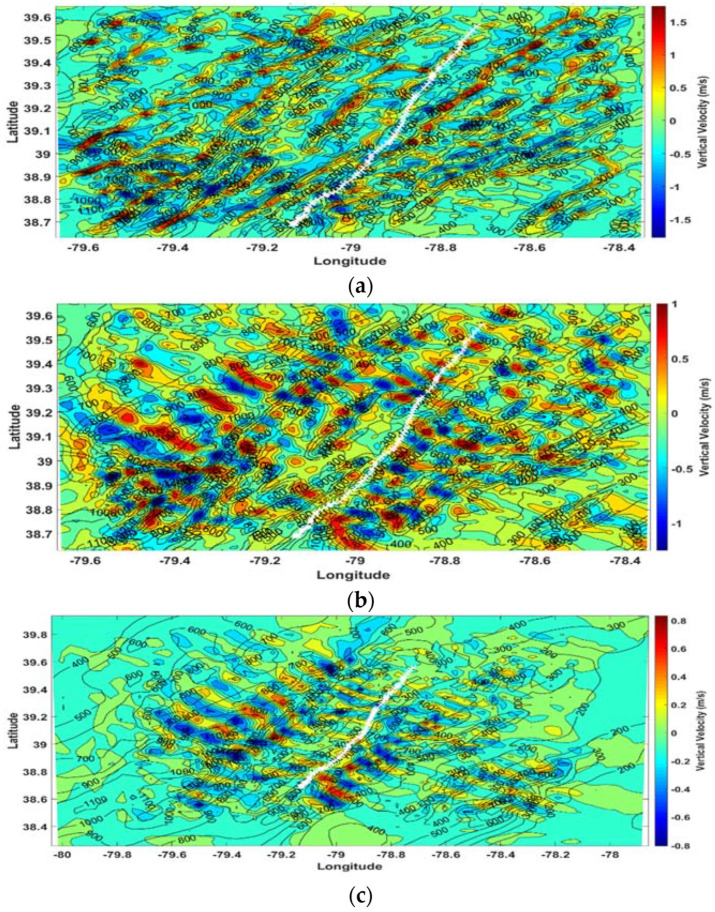
WRF model vertical velocity (m s^−1^) at 17 UTC along with location of the eagle flight track (white) for (**a**) Domain 3 at Level 15 (~250 m AGL); (**b**) Domain 3 at Level 30 (~1750 m AGL); and (**c**) Domain 2 at Level 30 (~1750 m AGL).

**Table 1 animals-12-01470-t001:** Weather Research and Forecasting (WRF) model specifications and parameterizations (NAMAN: North American analysis; RRTMG: rapid radiative transfer model for general circulation models; Noah: community land surface model; PBL: planetary boundary layer; MYJ: Mellor–Yamada–Janjic).

WRF Scheme/Feature	WRF Version 3.7.1
Time	Run at 12 UTC
Boundary Conditions	NAMANL
Horizontal Resolution	Domain 1: 9 km
	Domain 2: 3 km
	Domain 3: 1 km
Vertical Coordinate	Terrain Following
Vertical Levels	40
Longwave Radiation	RRTMG
Shortwave Radiation	RRTMG
Cumulus Parameterization	Kain–Fritsch
Microphysics	Morrison
Surface Layer	Eta
Land Surface Model	Noah
PBL Scheme	MYJ

**Table 2 animals-12-01470-t002:** Summary of small-scale flight maneuvers used during flight (CC: counterclockwise; and C: clockwise). See Appendix A for more detailed statistics.

	CC Circle	C Circle	Meander	Glide Up	Glide Down
Number of classified segments	44	42	8	10	17
Average segment time (s)	23.2	20.5	42.1	35	267.7
Average altitude change (m)	38.7	36.3	50.1	40	−282
Average climb/descent rate (m/s)	1.7	1.8	1.7	1.4	−1
Average ground speed (m/s)	n/a	n/a	n/a	15.7	19.1
Total time in maneuver (min)	17	14.4	5.6	5.8	75.8
Percent total time *	14.0	11.9	4.6	4.8	62.6
Net altitude change for maneuver (m)	1702.8	1524.6	400.8	400.0	−4794.0

* Not including missing or unclassified segments totaling 7.7 min.

**Table 3 animals-12-01470-t003:** Parameters and results of analysis of bird’s conversion of altitude gain into progress glide distance based upon the bird’s lift (L)/drag (D) ratio.

Parameter	Value	Source
Height gain during circling (H)	3227 m	Table 2
Time circling (T_c_)	32 min	Table 2
Time down gliding (T_g_)	76 min	Table 2
Tail wind speed around 1300 m AGL	11.0 ms^−1^	Appendix A
Average ground speed during glides	19.1 ms^−1^	Table 2
Lift/drag ratio (L/D)	15:1	Taylor et al., 2016
D_c_ is progress distance during circling	21.1 km	Calculated
D_g_ is progress distance during gliding	87.1 km	Calculated
D_t_ is the total distance flown in ~2 h	103.5.0 km	Measured

## Data Availability

The WRF-model output data, GPS flight data, and analytical methodologies (code) used in this study are available at http://www.datadryad.org (accessed on 13 May 2022).

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
