# Peer review of "An Instrumented Golden Eagle’s (Aquila chrysaetos) Long-Distance Flight Behavior"

_animals, 2022, doi:10.3390/ani12111470_

Round 1

Reviewer 1 Report

Although the idea is a good one and the results are interesting, I think the authors should consider expanding the sample both in terms of number of individuals and time analysed.

Having a very precise system that allows these measurements, the authors could have analyzed more hours and not only 2 hours of flight of a single individual. In order to fully understand the use of atmospheric waves by a bird, it would be necessary to carry out the same study with more individuals.

L-54 Can you reference any citations that demonstrate that the vertical velocities in the atmosphere decay rapidly with the height?

L-71. Authors show results when they should specify the objectives or hypotheses of their study. 

L-78. For this type of study, I consider that having a sample of only one individual is too small. I am sure that the authors could expand the sample to verify that the circumstance they are studying occurs in more than one individual golden eagle.

L- 90. The analyzed segment shows what the authors want to describe in the study, but again I consider the sample to be weak. It would be necessary to analyze more segments of the eagle's flight.

L90 I think the authors should tag more than one Golden Eagle and check their behavior 

L-133 ¿ What is the error in the GPS locations?

L-144 Although the condor and the eagle are raptors, they are different species and different families. So assuming that the eagle migrates without flapping its wings. Ideally, this assumption should be supported by data from the same species or family.

L-101 AGL is mentioned, but it is not explained what the acronym means until line 202.

L-202 Did you have to transform the flight height given by the GPS to get the height above ground level? If so, what procedure did you follow?

L-233 Do you have a method of calculating metabolic energy expenditure? If so, this should be given more importance.

L-261 Again, I insist again that all this could be checked by enlarging the sample and adding flight segments occurring in late winter/early spring: late winter/early spring.

Reviewer 2 Report

r.56: here atmospheric gravity waves are introduced for the first time, and a  reader might expect a more detailed description of this phenomenon

r.65: “most of these studies” is undefined, you might specify which of them already used GPS systems

r.265-267: I’m not convinced that “Alternating clockwise and counterclockwise circling …. is further evidence supporting slab-like vertical velocity zones typical of waves over the more toroidal organization typical of thermals” and that “Successive circling with reversing directions of circles were incompatible with the use of thermals” (r.319-320) . When a glider meets a thermal on its right side it would exploit it more efficiently by turning right towards the thermal’s core and therefore circling clockwise, while a thermal met from the glider’s left side should invite a counter-clockwise circling.  
